# Identifying Protein Features and Pathways Responsible for Toxicity Using Machine Learning and Tox21: Implications for Predictive Toxicology

**DOI:** 10.3390/molecules27093021

**Published:** 2022-05-08

**Authors:** Lama Moukheiber, William Mangione, Mira Moukheiber, Saeed Maleki, Zackary Falls, Mingchen Gao, Ram Samudrala

**Affiliations:** 1Department of Computer Science and Engineering, University at Buffalo, Buffalo, NY 14260, USA; lamamouk@buffalo.edu; 2Department of Biomedical Informatics, Jacobs School of Medicine and Biomedical Sciences, University at Buffalo, Buffalo, NY 14260, USA; wmangion@buffalo.edu (W.M.); miramouk@buffalo.edu (M.M.); zmfalls@buffalo.edu (Z.F.); 3Department of Mechanical Engineering, University at Buffalo, Buffalo, NY 14260, USA; saeedmal@buffalo.edu

**Keywords:** machine learning, random forest, feature selection, structure–activity relationships, high-throughput screening, enrichment analysis, proteomic signature, toxicity, drug behavior

## Abstract

Humans are exposed to numerous compounds daily, some of which have adverse effects on health. Computational approaches for modeling toxicological data in conjunction with machine learning algorithms have gained popularity over the last few years. Machine learning approaches have been used to predict toxicity-related biological activities using chemical structure descriptors. However, toxicity-related proteomic features have not been fully investigated. In this study, we construct a computational pipeline using machine learning models for predicting the most important protein features responsible for the toxicity of compounds taken from the Tox21 dataset that is implemented within the multiscale Computational Analysis of Novel Drug Opportunities (CANDO) therapeutic discovery platform. Tox21 is a highly imbalanced dataset consisting of twelve in vitro assays, seven from the nuclear receptor (NR) signaling pathway and five from the stress response (SR) pathway, for more than 10,000 compounds. For the machine learning model, we employed a random forest with the combination of Synthetic Minority Oversampling Technique (SMOTE) and the Edited Nearest Neighbor (ENN) method (SMOTE+ENN), which is a resampling method to balance the activity class distribution. Within the NR and SR pathways, the activity of the aryl hydrocarbon receptor (NR-AhR) and the mitochondrial membrane potential (SR-MMP) were two of the top-performing twelve toxicity endpoints with AUCROCs of 0.90 and 0.92, respectively. The top extracted features for evaluating compound toxicity were analyzed for enrichment to highlight the implicated biological pathways and proteins. We validated our enrichment results for the activity of the AhR using a thorough literature search. Our case study showed that the selected enriched pathways and proteins from our computational pipeline are not only correlated with AhR toxicity but also form a cascading upstream/downstream arrangement. Our work elucidates significant relationships between protein and compound interactions computed using CANDO and the associated biological pathways to which the proteins belong for twelve toxicity endpoints. This novel study uses machine learning not only to predict and understand toxicity but also elucidates therapeutic mechanisms at a proteomic level for a variety of toxicity endpoints.

## 1. Introduction

Exposure to persistent natural and synthetic environmental pollutants continues to be a significant health concern [1]. With technological advances in computational toxicology, systems biology, and bioinformatics, researchers and regulators have access to tools that allow for rapid assessment of toxic compounds, reducing the use of low-throughput, expensive, and time-consuming in vivo animal testing [2,3]. High-throughput screening (HTS) [4,5] has been utilized in conjunction with computational models to profile compounds for potential adverse effects and assess how compounds interact with biological systems. Furthermore, in the past few years, quantitative high-throughput screening (qHTS) has emerged as a powerful tool to allow the study of toxicological pathways linked with toxicity endpoints [6].

The Toxicology in the 21st Century (Tox21) program has been established as a collaborative effort among federal entities, including the National Center for Advancing Translational Sciences (NCATS), the National Institute of Environmental Health Sciences (NIEHS), the Environmental Protection Agency (EPA), and the Food and Drug Administration (FDA), to advance toxicity assessment. This program has applied qHTS to profile a library of around 10,000 compounds, including but not limited to environmental hazards, industrial chemicals, drugs, and food additives [7,8,9,10]. The Tox21 compound library was run against a panel of seven nuclear receptors (NR) and five stress response (SR) pathway assays, generating the most significant high-quality in vitro toxicity data to date [11]. Data generated from Tox21 has been used to identify compounds that interact with specific toxic pathways, including some not previously known [12,13,14,15]. This data has also been used, together with chemical structure information, to train predictive machine learning models to compute whether a chemical will elicit a particular toxicological outcome based on in vitro findings [16,17,18,19,20,21]. Deep learning has achieved the best prediction performance in toxicity prediction on the Tox21 data [22]. However, one major limitation of deep neural networks is that they tend to mask the effective features for toxicity prediction.

We propose a novel approach to building a pipeline that predicts the essential proteins implicated in in vivo toxicity, which are then fed into an enrichment analysis to assess the mechanistic pathways contributing to each toxicity endpoint and provide a high-level biological interpretation. To extract important protein feature descriptors which contribute most to the target prediction, many feature selection techniques originating from machine learning have been proposed [23,24,25,26]. Random forest [27], which is an embedded feature selection technique, has emerged as an efficient and robust algorithm that can address feature selection, even with a higher number of variables [28,29,30]. The random forest approach constructs many decision trees during training and averages the predicted values to obtain the final outputs. Due to the random exploration of features at each node in the tree construction, random forest lends itself to feature selection. Determining and assessing the most relevant feature descriptors between compounds and toxicity endpoints on a mechanistic and proteomic scale will enable us to comprehensively evaluate compound toxicity.

Previous studies using machine learning models on Tox21 data have shown that the data is highly imbalanced, with a greater proportion of inactive or non-toxic compounds for each toxicity assay than active ones [31,32,33]. Highly imbalanced data can be a problem when training machine learning classification models, as the model becomes biased toward the inactive class, resulting in a higher misclassification rate for the active class. Most of the previous classification algorithms using Tox21 data have not handled the imbalanced problem for toxicity prediction explicitly [31,32,34,35]. We propose an improved random forest method for feature selection to find the relevant features for toxicity prediction, i.e., random forest with the combination of the Synthetic Minority Oversampling Technique (SMOTE) and the Edited Nearest Neighbor (ENN) method, aka SMOTE+ENN, a resampling method to handle the imbalanced problem.

The Tox21 dataset has been well-studied using chemical descriptors as features to understand toxicity [16,33]. In contrast, this study utilizes the Computational Analysis of Novel Drug Opportunities (CANDO) platform to obtain the protein feature descriptors to understand toxicity at the protein pathway level. CANDO is a multiscale shotgun drug discovery, repurposing, and design platform which employs multitargeting to generate proteomic scale interaction signatures for any small molecule, including approved drugs, against large libraries of protein structures from various organisms [36,37,38,39,40,41,42,43,44,45,46,47,48,49]. The proteomic interaction signatures are analyzed to computationally assess compound similarity, with the premise that drugs with similar signatures may treat the same diseases. Version 3 of the platform now features several protein-related and drug-related biological entities, such as protein pathways, protein-protein interactions, protein-disease associations, and adverse drug reactions, further enhancing the ability to compare small molecule compounds in the context of biological systems. CANDO has been validated in multiple indications with an overall success rate of 58/163, not including 51 drug candidates with activity against SARS-CoV-2 (out of 275 ranked predictions) extracted from in vitro and electronic health record-based studies published in the literature [46]. However, CANDO has yet to be employed explicitly to predict compound toxicity, despite how conducive this multiscale framework is for that task.

Our study aims to improve toxicity prediction by combining the capabilities of machine learning, the CANDO platform, and enrichment analysis to identify the most effective protein structures for predicting twelve toxicity endpoints from the Tox21 data and highlighting implicated biological pathways. We begin the study by pre-processing the data, generating protein feature descriptors using CANDO, and performing dataset balancing. Following this, we apply the random forest algorithm to select the top structural protein descriptors and subject them to enrichment analysis to identify high-level biological entities that explain the mechanisms through which the toxicity is induced. We provide a case study of aryl hydrocarbon receptor (AhR) activation, comprehensively identifying the pathways potentially responsible for its associated toxicity using only those identified via enrichment analysis. Our approach of combining CANDO, machine learning, and feature selection allows for a detailed understanding of compound behavior and a greater ability to predict not only toxicity but also mechanistic etiology.

## 2. Results and Discussion

In this section, we present (1) a summary of the curated and prepossessed Tox21 datasets; (2) performance metrics of our computational algorithm on the Tox21 data; (3) a comparison between our study and other published Tox21 studies; (4) the enriched pathways for the twelve Tox21 assays; and (5) a case study of NR-AhR toxicity with published literature supporting our computational results and enrichment analysis. Figure 1 provides a high-level overview of our study design (Section 4).

### 2.1. Preprocessed Data

Table 1 shows the preprocessed Tox21 compounds and their activities measured by twelve qHTS in vitro assays. The number of active and inactive compounds for each of the twelve qHTS assays was computed, along with the imbalance ratio, which is the ratio of the majority class (inactive non-toxic compounds) to the minority class (active toxic compounds). The imbalance ratio varied greatly between the twelve assays, with the inactive compounds being the predominant majority (ratio 10:1 or higher) compared to the actives. The higher the imbalance ratio value, the more imbalanced the activity class distribution for the assay. In the training datasets, the imbalance ratio ranged from 5.52 for the SR mitochondrial membrane potential (SR-MMP) assay up to 37.62 for the NR peroxisome proliferator-activated receptor γ (NR-PPAR-γ) assay. The test datasets had imbalanced ratios equivalent to or larger than those of their corresponding training datasets, ranging from 5.06 for the SR antioxidant response element (SR-ARE) assay up to 69.75 for the NR androgen receptor ligand-binding domain (NR-AR-LBD) assay.

### 2.2. Comparison to Other Tox21 Studies

Our study used the random forest classifier to avoid overfitting and enhance performance. Further, it was a commonly used model by participating teams in the Tox21 Data Challenge. Two of the winning teams used a random forest model to achieve the best performance in predicting compound toxicity against the NR-androgen receptor (NR-AR), NR-aromatase, SR-p53 [20], and NR estrogen receptor alpha ligand-binding domain (NR-ER-LBD) assay [19]. The area under the receiver operating characteristic curve (AUCROC) and the winning teams’ balanced accuracy scores were provided as evaluation metrics during the Tox21 Data Challenge.

Banerjee et al. [32] highlighted the importance of using sampling methods when training a classifier on imbalanced chemical data, such as the Tox21 dataset. This is important because non-sampling methods on imbalanced data result in poor recall performance due to the classifier favoring the majority inactive class [32]. Another study on a similar methodology employed a random forest classifier with different resampling techniques and showed that the random forest with the SMOTE+ENN classifier performed the best on the Tox21 data [31]. Therefore, in this study, we implemented the random forest and SMOTE+ENN algorithms to train our model for feature selection and handle class imbalance. The SMOTE+ENN classifier was applied to resample the training data, and the random forest classifier was then fit to the training data. The trained model was used to make predictions on the test data and was evaluated using classification performance metrics. This approach was applied to each of the twelve assay datasets.

Table 2 reports the model evaluation metrics for the random forest classification model for twelve qHTS assays. The reported values varied depending on metrics and assays. The assays with a more favorable imbalance ratio perform much better. Overall, our modeling approach achieved decent performance measured by AUCROC ≥ 0.7, except for NR-AR-LBD, which had very few active labels, as shown by its high imbalance ratio of 30.9 in the training set. The AUCROC has the highest value of 0.92 for the SR-MMP assay, with the lowest imbalance ratio of 5.52 in the training set. This is not surprising as the greater the number of active labels, the better the model’s discriminatory power between the active and inactive classes. The SR-MMP assay also had the highest F1-score of 0.49 and the highest recall and high precision. The area under the precision-recall curve (AUPRC) score was also high for SR-MMP, with a score of 0.60, signifying that the model could handle the positive samples correctly.

We compared the performance of our approach with those employed in the Dmlab [20], and Microsomes [19], which are the top two performing random forest models in the Tox21 Data Challenge. We also compared our results with random forest with SMOTEENN (SMN), which was the best random forest-based classifier employed by Idakwo et al. [31]. Since AUCROC and balanced accuracy are the only accuracy measures in the Tox21 Data Challenge, we only evaluated these two metrics in our results. Figure 2 depicts a comparison of the model performance in terms of AUCROC and the balanced accuracy of the other studies and our study. On average, our model performed comparably to the Dmlab, Microsomes, and SMN model approaches based on the AUCROC and balanced accuracy metrics across most of the twelve assays. Yet our model allows for interpretability, as it enables the extraction of important proteins implicated in each toxicity assay.

Despite the performance similarity with other studies, it is not a direct comparison since our algorithm was used for feature selection, while the compared random forest-based models are employed for prediction. Furthermore, such a comparison requires using the same data pre-processing methodology and feature descriptors when training a model, which was not done in this case.

### 2.3. Enrichment Analysis

The top 100 most important protein features as determined by the random forest algorithm for each Tox21 endpoint were subjected to enrichment analysis in the context of protein pathways. The average number of enriched entities obtained for the twelve toxicity endpoints was 36, with the minimum being 13 for NR-PPAR-γ and the maximum being 59 for SR-ATAD5. The exact proteins and pathways highlighted for the NR-AhR activation assay are provided in Table 3, including the total number of proteins in the pathway and the calculated *p*-value. The full names and UniProt identifiers of those genes are available in Appendix A.

### 2.4. Case Study for NR-AhR

We selected the NR-AhR endpoint as a case study due to its model performance relative to other NR assays. AhR is a helix-loop-helix ligand-activated transcription factor that binds a wide range of ligands, including environmental pollutants, such as polycyclic aromatic hydrocarbons (PAHs) and polyhalogenated aromatic hydrocarbons [51]. The latter class of compounds includes halogenated dibenzo-p-dioxins, also known as dioxins. One of the dioxin compounds is 2,3,7,8-tetrachlorodibenzo-p-dioxin (TCDD), which is the most commonly used environmental agent for studying AhR-mediated biochemical and toxic responses because of its high affinity to AhR [52]. It also causes a wide range of toxic effects, including immunosuppression and tumor promotion [53,54]. Previous studies have suggested that AhR signaling elicits numerous critical biological processes, including the modification of the cell cycle, cell proliferation, immune responses, and tumorigenesis [55,56,57,58]. Toxicity induced via AhR activation arises via genomic and non-genomic signaling pathways (Figure 3). The genomic signaling pathway involves the ligand-activated AhR translocating in the nucleus and dimerizing with AhR nuclear translocator (ARTN). The activated AhR/ARNT heterodimer complex interacts directly or indirectly with DNA by binding to recognition sequences located in the promoter regions of dioxin-responsive genes. This leads to adverse changes in cellular processes, leading to the toxic response [59,60]. The non-genomic signaling is due to the suppression or activation of certain enzymes via the ligand-activated AhR [61].

We assessed the interpretability of our pipeline by analyzing the NR-AR selected proteins and enriched pathways to assess the toxic response. Based on our enrichment analysis, the AhR signaling pathway is involved in 34 different toxic signaling pathways. We analyzed a handful of pathways, including their corresponding proteins to illustrate the validity of our toxicity analysis.

Several studies have shown that ligand-activated AhR is involved in toxicity and cancer via DNA damage and cell cycle disruption. However, the molecular signaling mechanism by which this occurs is unknown [63]. It is hypothesized that TCDD exposure causes cancer by affecting the repair of double-strand breaks mediated by the AhR signaling pathway [63]. Further, Rattenborg et al. showed that TCDD alters the expression of the tumor suppressor gene breast cancer type 1 susceptibility protein (BRCA1) by downregulating BRCA1 promoter activity. However, the exact mechanism by which TCDD suppresses BRCA1 activity is unclear. [64]. BRCA1, which contains a E3 ubiquitin-protein ligase domain, plays a central role in DNA repair, cell cycle control, and tumor suppression [65,66,67]. The schematic representation of the BRCA1 pathway is depicted in Figure 4. A similar study in the literature found that benzo[a]pyrene (BaP), a polycyclic AhR ligand [68], exerts its carcinogenicity by inhibiting BRCA1. These studies indicate the participation of the AhR pathway and its effects on DNA damage and cancer but do not mention the subsequent downstream signaling pathway and protein regulation affected by the ligand-activated AhR complex interaction. Furthermore, a study by Foo et al. highlighted that disruptions to either the ataxia-telangiectasia mutated (ATM)-mediated phosphorylation and the cell cycle G2-M checkpoint impact BRCA1-related cancers [69]. This supports our analysis as our pipeline selected the ATM-mediated phosphorylation and the G2-M checkpoint pathways as being involved in AhR toxicity, leading to the hypothesis that AhR ligands (BaP and TCDD) interact with BRAC1, modulating these pathways.

Our pipeline selected BRCA1 as one of the important proteins in causing AhR toxicity, corroborated by the research studies mentioned previously. This indicates that the toxic AhR compounds in the Tox21 library cause carcinogenicity and DNA damage by binding and modulating the activity of BRCA1. Further, we identified specific DNA damage mechanisms (pathways and proteins) mediated by BRCA1, including the DNA double-strand break response, processing of DNA double-strand break ends, nonhomologous end-joining, ATM-mediated phosphorylation, and the G2-M checkpoint, along with other proteins that play an essential role in DNA damage signaling and cell cycle control, such as RNF8, UBE2N, RAP80 and NSD2, RPA1.

As previously mentioned, AhR is an important factor that regulates immune responses. Several studies have implicated that the AhR signaling pathway interacts in autoimmune and inflammatory diseases. BaP, an AhR ligand that is a prominent carcinogenic component of cigarette smoke, smog, and some over-cooked foods [70], has been shown to exert its toxicity by inhibiting osteoclastogenesis through the activation of the nuclear factor-κB (NF-κB) pathway [71]. The NF-κB pathway is known to be involved in inflammatory arthritis and osteoclastogenesis [72,73]. The TNF receptor-associated factor 6 (TRAF6), a protein selected by our pipeline as involved in toxicity, is an essential factor for osteoclastogenesis. It leads to the activation of the NF-κB pathway [74]. In addition to BaP, another AhR ligand that contributes to inflammation is TCDD. TCDD has been shown to contribute to rheumatoid arthritis, a chronic autoimmune disease that causes joint inflammation, bone destruction, and increased pro-inflammatory cytokines, which is modulated by the NF-κB pathway [75].

Further, a study in the literature demonstrated that TCDD activates the p38–MAPK pathway, implying a link between p38-MAPK signaling and AhR. It demonstrated that the AhR and p38–mitogen-activated protein kinase (MAPK) induce the expression and activity of c-Jun, a proto-oncogene [76], highlighting a novel mode of interaction between the AhR and p38–MAPK pathway in carcinogenicity. Our algorithm selected the p38–MAPK-dependent pathway as important for AhR toxicity including JNK (c-Jun kinases) phosphorylation and activation mediated by activated human TAK1 and activated TAK1 mediates p38 MAPK activation. Our pipeline selected interconnected pathways, including MAPK and NF-κB signaling, depicted in Figure 5, highlighting that when compounds bind to the proteins in these pathways and subsequently modulate their proper signaling, the outcome is the observed toxic phenotypes.

The NF-κB signaling and MAPK pathway are illustrated in Figure 5. One way the NF-κB and mitogen-activated protein kinase (MAPK) pathways are activated is through the interleukin-1 receptor/toll-like receptor L-1R/TLR) and TLR7/8/9. Once IL-1R/TLR or TLR7/8/9 is activated, it recruits the myeloid differentiation primary response 88 (MyD88). MyD88 recruits interleukin-1 receptor-associated kinases (IRKA1/4). The IRKA1/4 complex then recruits TRAF6 protein. Once TRAF6 is activated, it is recruited by UBE2N/UBC13, which are ubiquitin-conjugating enzymes that catalyze the synthesis of a lysine-63-linked polyubiquitin chain [77]. TRAF6 activates the TAK1 molecule, which activates several downstream molecules. It activates the I-κB kinase (IKK) complex, leading to NF-κB pathway activation. TAK1 can also activate MAPKs (P38 and c-Jun kinases (JNK)), which in turn activates the AP-1 transcription factor. AP-1 enters the nucleus and causes the transcription of pro-inflammatory cytokines. In addition to activating TAK1, TRAF6 can also activate downstream interferon regulatory factor 7 (IRF7) via the TLR7/8/9-MyD88 pathway, leading to the production of type 1 interferons.

Another way the NF-κB pathway is activated is through the TNFR1 receptor. Once the TNFR1 receptor is activated, the TRADD protein binds to a domain on the TNFR1 receptor. Then, the TNF receptor-associated factor 2 (TRAF2) and receptor-interacting protein 1 (RIP1) kinase are recruited. The RIP1 protein ultimately activates the IKK complex. The IKK complex consists of IKKα/IKKβ kinases and NF-κB essential modulator (NEMO). Then IKK phosphorylates the Ikb protein, which masks the nuclear localization signal on NF-κB. The phosphorylation of the Ikb protein causes its degradation by proteasomes, allowing NF-κB to translocate into the nucleus. Once in the nucleus, NF-κB binds to genes such as pro-inflammatory genes leading to cytokine production.

TRAF6 was selected by our pipeline as an essential protein in predicting AhR toxicity along with others such as UBE2N and TRAF2. A recent study highlighted that the interaction between TRAF6 and UBE2N is crucial to the NF-κB inflammatory pathway [78]. This signifies that one of the mechanisms by which AhR toxicity is induced is via modulating proteins leading to autoimmune inflammation. Further, our pipeline identified pathways connected with or upstream of the NF-κB pathway, including TRAF6-mediated NF-κB activation, TRAF6-mediated IRF7 activation, IRAK1 recruitment of the IKK complex, IRAK1 recruitment of the IKK complex upon TLR7/8 or 9 stimulation, TRAF6-mediated IRF7 activation in TLR7/8 or 9 signaling, RIP1-mediated IKK complex recruitment, and TAK1-activated NF-κB by phosphorylation and activation of IKKs complex. These multiple interconnected pathways are depicted in Figure 5.

Our AhR toxicity analysis shows that our algorithm selected proteins and their corresponding pathways that are corroborated by literature analysis. In addition, these selected proteins overlap across multiple pathways, showing that toxicity is not due to single target binding but rather a complex interconnected network involving many pathways. The overlap in proteins and pathways indicates that our algorithm can potentially provide novel insight into understanding toxicity from a proteomic and pathway perspective. However, further prospective studies are warranted to elucidate the importance of these pathways in AhR toxicity and the other toxicity endpoints.

## 3. Limitations and Future Work

While several studies in the literature have used chemical properties as feature descriptors to predict toxicity [16,17,18,19,20,21], this work is, to our knowledge, the first to use protein descriptors for feature selection. In this study, we looked at toxicity not only from a single protein and target-binding perspective but also in the context of pathways and signaling events. This study is useful to investigate the toxicity targets and pathways interconnected with well-known toxicity pathways.

Examining the broad spectrum of possible pathways enriched for a specific toxicity endpoint enables the investigation of multiple proteins and target binding sites leading to the corresponding toxicity. Our literature-based validation demonstrated that our computational pipeline predicts pathways and proteins associated with AhR toxicity. Even though some experimental studies from the literature support our predictions, further prospective experimental studies are warranted to unequivocally confirm their association with AhR toxicity. These areas are worth improving on in the future by implementing novel machine learning feature selection techniques. In summary, our pipeline implemented within the CANDO platform is useful for hypothesis generation and elucidating toxicity mechanisms at proteomic and pathway scales.

## 4. Materials and Methods

### 4.1. Tox21 Datasets

The Tox21 compound structures and activity measurements for twelve different qHTS assays were extracted from the Tox21 Data Challenge [11]. The training, evaluation, and test datasets consisted of 11,764, 296, and 647 compounds, respectively. We combined the training and evaluation datasets to form our final training data and used the test data for model evaluation. The twelve qHTS in vitro assays consisted of two categories, seven of which were part of the nuclear receptor (NR) and five part of the stress response (SR) pathways. The NR assays included the androgen receptor (AR), androgen receptor ligand-binding domain (AR-LBD), aryl hydrocarbon receptor (AhR), aromatase, estrogen receptor (ER), estrogen receptor luciferase assay (ER-LBD), and peroxisome proliferator-activated receptor γ (PPAR-γ). The SR assays included the antioxidant response element (ARE), heat shock factor response element (HSE), p53, mitochondrial membrane potential (MMP), and ATPase Family AAA domain containing 5 (ATAD5). In each assay, the activity of a compound was assigned a class label, where a label of 1 signified that the compound was active, i.e., toxic, and a label of 0 signified that the compound was inactive, i.e., not toxic. There were duplicates and inconsistent activity labels for the compounds across the twelve assays (see Section 4).

### 4.2. UniProt

A human protein library of 19,582 sequences was extracted from UniProt [79]. Of these, 4966 had at least one solved X-ray diffraction structure in the Protein Data Bank (PDB) [80], including 4641 with one chain, 298 with two chains, and 27 with three chains, totaling 5316 total human structures after removing two non-viable ones. These protein structures were chosen by matching their UniProt IDs to all corresponding structures in the PDB, filtering for chains with the most significant sequence coverage to the whole sequence, and then selecting the chain with the best resolution. Proteins were mapped to 2219 pathways in Reactome [81] with an average of 35.1 structures per pathway.

### 4.3. Protein-Compound Interaction Scoring Protocol

The Computational Analysis of Novel Drug Opportunities (CANDO) therapeutic discovery, repurposing, and design platform [36,37,38,39,40,41,42,43,44,45,46,47,48,49] was used to generate protein interaction signatures for every molecule in its drug/compound library, which served as the feature extraction section in our pipeline. These protein interaction signatures were used as features in our machine learning development. The interaction scores between every compound in the Tox21 dataset and all structures in the human protein library were computed using a rapid in-house bioanalytical docking (BANDOCK) protocol [43]. First, binding sites were predicted for each protein using COACH [82], a consensus method combining structural and sequence similarity to proteins in the PDB [80], with each prediction having a confidence score (BScore) as well as an associated co-crystallized ligand. Depending on the scoring protocol used, the output interaction score for the compound and protein considers both the BScore and the molecular fingerprint similarity between the compound and the associated ligand (CScore). In this study, the scores were determined by multiplying the BScore by the CScore, which itself is the Sorenson–Dice coefficient [83] that measures the similarity between the ECFP4 fingerprints (computed using RDKit [84]) of the query compound and binding site ligand. Since the COACH algorithm outputs multiple binding sites (and therefore associated ligands) for each protein, the maximum value of the product of BScore and CScore is the chosen interaction score; this serves as a measure of the likelihood of interaction between a compound and a protein, i.e., a proxy for binding strength.

### 4.4. Study Design

A general overview of the workflow of this study is illustrated in Figure 1. Our study design consisted of data pre-processing, feature generation, resampling, feature selection, and enrichment analysis. Data pre-processing and feature generation was implemented on both the training and test datasets. Resampling using SMOTE+ENN and feature selection using the random forest algorithm was applied separately to each of the twelve assays in the training dataset. This was followed by model optimization using a repeated stratified three-fold cross-validation using the training data and model evaluation using the test data. Enrichment analysis was conducted on the ranked list of top 100 protein descriptors selected by the algorithm to analyze the Tox21 toxicity assays at a proteomic and pathway level. These steps are described in detail in the subsequent sections.

#### 4.4.1. Data Pre-Processing and Feature Generation

Compound structures were extracted from the Tox21 Data challenge as SMILES strings. Compound structure standardization and normalization were implemented using the RDKit MolVS library [50]. We applied a fragmentation step as described in [31], where SMILES with salt moieties, varied resonance structures, and tautomers were removed, and the valid SMILES were canonicalized by normalizing inconsistent chemical groups. To handle the ambiguous compounds with duplicate activities (0 or 1) for the same toxicity assay, we removed compounds with an equal number of active and inactive labels for that particular target. For compounds with an unequal number of active and inactive labels, the most frequent activity label was selected.

Following the normalization and merging of the compounds, the resulting SMILES were used to generate a compound-protein interaction score matrix using the BANDOCK interaction scoring protocol [43]. This provided the feature descriptor input to the model. The matrix consisted of 7808 compounds for the training set and 628 compounds for the test set. The generated matrix consisted of 8385 proteins as feature descriptors. The feature space was filtered for interactions with only solved protein structures, leading to 5316 proteins. The elimination of features reduced the computational requirement and the effect of the curse of dimensionality [85]. The resulting CANDO matrix and the activity measurements of the compounds for each of the twelve different qHTS assays were merged, generating twelve training datasets, one per assay. For each assay, only compounds labeled active or inactive were retained.

To understand the activity measurement distribution for each of the twelve assays, the number of active and inactive compounds for each of the twelve assays was computed, along with the imbalanced ratio, which is the ratio of the majority class (inactive non-toxic compounds) to the minority class (active toxic compounds).

#### 4.4.2. Data Resampling for Predictive Modeling

As mentioned before, the sparsity in the Tox21 data results in a model that favors the predominant inactive class and misclassifies the minority active class, which is usually the class of interest. We balanced the class distribution for each training set by implementing the SMOTE+ENN method to address this challenge. SMOTE+ENN combines both oversampling (using SMOTE) and undersampling (using ENN). SMOTE [86] synthesizes samples in the minority class by linear interpolation to increase the number of instances in the minority class. ENN reduces the number of instances in the majority class by removing noisy samples from the majority class, which is inconsistent with its k-nearest neighbors [87]. The SMOTE+ENN algorithm has been shown to deliver promising results when applied to imbalanced datasets with a small number of positive instances, including the Tox21 data [31,88].

#### 4.4.3. Random Forest

Protocols based on random forest have gained popularity in computational biology research owing to their unique advantages as being non-parametric, interpretable, and highly accurate for cheminformatic modeling [89,90]. Previous studies using Tox21 have demonstrated the utility of random forest in predicting compound toxicity [19,20,21].

Random forest is an ensemble learning method combining decision trees as base learners for increased performance [27]. Each tree is trained by different bootstrap samples having the same size as the training set. By selecting a random subset of features at each node in the tree construction, random forest introduces randomization and increased diversity in the forest, reducing the variance of the base learners [91]. The construction of random forest is described by the following steps:(1)Draw a bootstrap sample: we randomly sample N compounds with replacement from the original dataset;(2)Create maximum decision trees: we construct a decision tree for each bootstrap sample by randomly sampling a subset of features at each node and choosing the best split among those features;(3)Construct a forest by repeating steps 1 and 2 for N trees;(4)Predict the outcome: from the built forest, the prediction is obtained by aggregating the predictions of the N trees (i.e., majority votes for classification and average for regression tasks).

Given that the Tox21 dataset is high dimensional with a large feature space, traditional methods can lead to model overfitting. However, the random forest algorithm is less susceptible to model overfitting due to the utilization of ensemble strategies and random sampling, and therefore, we selected it as our modeling algorithm for this study [30].

#### 4.4.4. Random Forest for Feature Selection

The high-dimensional feature space of many tasks in bioinformatics has created a need for sophisticated feature selection techniques [92]. Feature importance is the process of finding parts of the input feature space relevant to a prediction task. The main advantage of using random forest compared to other machine learning algorithms is that it directly performs feature selection during model construction.

A commonly used measure to evaluate the importance of each predictor variable from a random forest classifier is the Gini importance measure, also known as the mean decrease of impurity. As a simple feature importance to rank features, random forest Gini feature importance has gained popularity in bioinformatics applications [92].

Gini importance is derived from the Gini index [27], which is a criterion used to determine which feature to split during tree training. It measures the level of impurity of the samples assigned to a node based on a split. Gini index ranges between 0 and 1, where 0 means all instances belong to one class, and 1 means that the instances are distributed randomly across the classes. The smaller the Gini index, the purer the node. The Gini index is measured as follows:(1)$$Gini\left(P\right)=\sum_{i=1}^{n}p_{i}\left(1-p_{i}\right)=1-\sum_{i=1}^{n}p_{i}^{2},$$
where $P=\left(p_{1},p_{2},\ldots ,p_{n}\right)$, $p_{i}$ is the probability of the class *i* at a certain node, and *n* is the number of classes [90].

The Gini importance value of a feature is computed as the sum of the Gini indices weighted by the probability of reaching that node averaged among all trees in the forest. A high Gini importance means that the feature is likely to be informative [93].

#### 4.4.5. Model Training and Testing

Following data pre-processing, SMOTE+ENN was applied to the training data to oversample the minority class and obtain an augmented training set to train the random forest. The random forest was trained with stratified ten-fold cross-validation with three repetitions to optimize the model’s hyper-parameters. The hyper-parameters that were optimized included the maximum depth of the tree and the number of trees in the forest. Other model hyper-parameters were set to their respective default values in scikit-learn [94]. Once the random forest model was optimized using the training data, the unseen test data was used to asses the model generalization performance and evaluate the effectiveness of the model. Afterward, the Gini importance was calculated to extract the top 100 weighted feature descriptors.

The model training was applied to each of the twelve datasets, generating an optimized feature selection model with the optimal feature descriptors per assay. The re-sampling technique was applied using the imbalanced-learn package in Python [95], and the random forest model was implemented via the scikit-learn Python library [94].

#### 4.4.6. Performance Evaluation Metrics

Once the model was optimized using the training data, the unseen test data was used to evaluate its performance. We reported both the area under the precision-recall curve (AUPRC) and the area under the receiver-operating characteristic curve (AUCROC) given the class imbalance for the task [96]. We also reported the accuracy metric (the proportion of correct predictions relative to the total number of compounds), with the caveat that high accuracy does not translate into a model that correctly predicts the rare inactive class and can therefore be misleading for evaluating model performance [97].

The decision made by the random forest classifier can be represented in a 2-by-2 confusion matrix, given that we have a binary classifier. The confusion matrix has four categories: true positives (TP), the number of active compounds that are correctly labeled; false positives, which is the number of incorrectly labeled inactive chemicals; true negatives, which refers to the number of correctly labeled inactive compounds; and lastly, false negatives (FN), which correspond to the number of incorrectly labeled inactive compounds.

We utilized imbalanced classification metrics derived from the confusion matrix, including recall, precision, F1-score, balanced accuracy, specificity, and Matthews’s correlation coefficient (MCC). Recall is a measure of the accuracy of the active minority class. It gives the proportion of actual positives identified correctly by the model, i.e., the number of correctly labeled active compounds out of the actual active compounds. Precision (also called positive predictive value) gives the proportion of positive identifications that were actually correct, i.e., the number of correctly labeled active compounds out of the compounds predicted as active. The F1-score is the harmonic mean of precision and recall. Specificity is a measure of the accuracy of the inactive majority class. It calculates the proportion of the actual negatives identified correctly by the model, i.e., the number of correctly labeled inactive compounds out of the actual inactive compounds. Balanced accuracy is the average of sensitivity and specificity and is useful when evaluating a classifier, especially when the classes are imbalanced. MCC is another metric used to assess the quality of binary classification for imbalanced data. It measures the correlation of the true classes with the predicted labels. MCC takes into account all of the four confusion matrix categories. These evaluation metrics were implemented using the scikit-learn package in Python [94]. The formulae of the evaluation metrics are as follows:(2)$$\mathrm{Recall}\;=\;\mathrm{Sensitivity}=\frac{\mathrm{TP}}{\mathrm{TP}\;+\;\mathrm{FN}}$$
(3)$$\mathrm{Precision}=\frac{\mathrm{TP}}{\mathrm{TP}\;+\;\mathrm{FP}}$$
(4)$$F1\text{-}\mathrm{score}=2\times \frac{\mathrm{Precision}\times \mathrm{Recall}}{\mathrm{Precision}\;+\;\mathrm{Recall}}$$
(5)$$\mathrm{Specificity}=\frac{\mathrm{TN}}{\mathrm{TN}\;+\;\mathrm{FP}}$$
(6)$$\mathrm{Balanced}\;\mathrm{accuracy}=\frac{\mathrm{Sensitivity}\;+\;\mathrm{Specificity}}{2}$$
(7)$$\mathrm{MCC}=\frac{\left(\mathrm{TP}\times \mathrm{TN})-(\mathrm{FP}\times \mathrm{FN}\right)}{\sqrt{\left(\mathrm{TP}\;+\;\mathrm{FP})\times (\mathrm{TP}\;+\;\mathrm{FN})\times (\mathrm{TN}\;+\;\mathrm{FP})\times (\mathrm{TN}\;+\;\mathrm{FN}\right)}}$$

#### 4.4.7. Enrichment Analysis

The top 100 most essential proteins for predicting each toxicity endpoint for the compounds in Tox21 dataset were inputs to an enrichment analysis protocol. The enrichment analysis identifies significantly overrepresented pathways based on their mappings to proteins obtained via Reactome. It utilizes the hypergeometric distribution to determine the probability that a pathway is significantly overrepresented based on the number of proteins in the top feature set associated with the pathway relative to the whole human proteome and the total number of proteins associated with it in that pathway. A *p*-value is computed using the probability mass function with the number of top features associated with the pathway serving as input. The human proteome used in this study includes 19,582 proteins from UniProt, of which 4966 have at least one solved X-ray diffraction structure available in the PDB and are mapped to at least one biological pathway.

## 5. Conclusions

To the best of our knowledge, this is the first computational pipeline that utilizes protein descriptors to extract the important features from the twelve toxicity endpoints in the Tox21 dataset to evaluate compound toxicity. We employed a combination of protocols within the CANDO drug discovery platform, including compound-proteome interaction signature generation, data balancing, feature selection, and enrichment analysis to understand compound toxicity behavior at the protein pathway level. We expect this computational pipeline will provide a novel perspective in evaluating environmental compounds and allow researchers and the pharmaceutical industry to explore the underlying proteomic mechanisms that not only induce toxicity but also potentially assist in developing novel therapeutics to mediate toxicity targets.

## Figures and Tables

**Figure 1 molecules-27-03021-f001:**
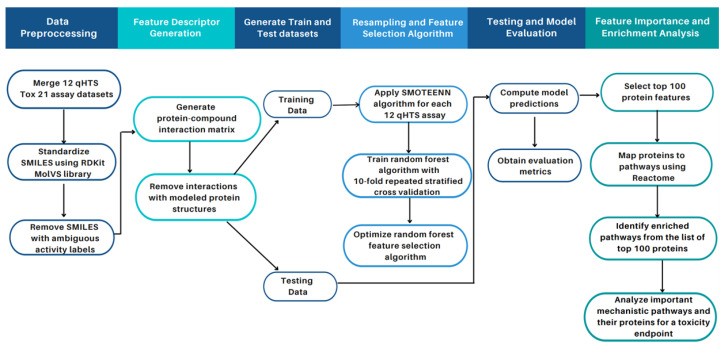
**Overview of study workflow and pipeline for toxicity feature identification.** As part of the data pre-processing step, the twelve Tox21 assay datasets containing the SMILES strings and activity class of the compounds were merged. The compounds were normalized and standardized using the MolVs library [50], built on RDKit where compound duplicates with ambiguous activity labels (i.e., equally active and inactive labels for the same compound) were removed. Model features were generated using a protein-compound interaction matrix via the CANDO platform. The data containing the features and the class activity for each compound were generated for each of the twelve assays and split into training and test sets. The SMOTE+ENN algorithm was applied to oversample the minority class and obtain an augmented training subset used to train our random forest classifier. The parameters for the random forest classifier were selected using tenfold cross-validation to attain optimal model performance. The model was evaluated on the unseen test data with metrics such as F1-score, recall, precision, specificity, balanced accuracy, AUCROC, and AUPRC to evaluate its performance. The model was then used to obtain the top 100 important features (protein structures); proteins were associated with pathways annotated in Reactome. The overrepresented pathways in the top 100 proteins via enrichment analysis were identified. The enriched pathways for the NR-AhR assay were analyzed as a case study, which illustrated that our unique proteomic feature selection pipeline allows for a mechanistic understanding of compound toxicity.

**Figure 2 molecules-27-03021-f002:**
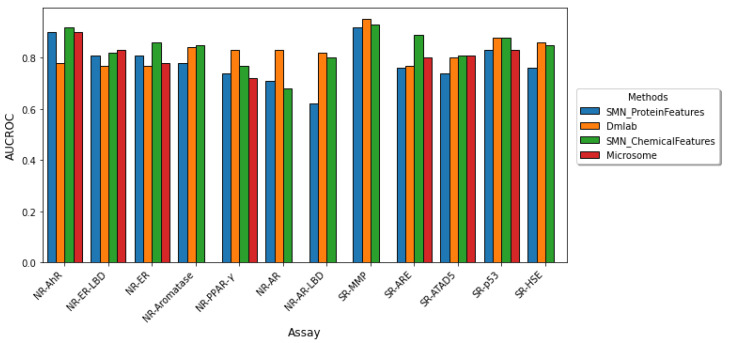
**Comparison of model performance across the twelve Tox21 assays.** The performance of the twelve Tox21 assays is evaluated using AUROC (**top**) and balanced accuracy (**bottom**). Although AUCROC is widely used as a binary classifier evaluation metric, it can be misleading for imbalanced classification with few examples of the minority class. We compared our results in blue to the three other studies which utilized a random forest-based classifier. (The microsome study does not report both AUCROC and BA for the NR-AR, NR-AR-LBD, and SR-MMP assays and the BA for NR-PPAR-γ.) Our model performs comparably to previous methods yet allows for the extraction of important protein features implicated in each toxicity endpoint.

**Figure 3 molecules-27-03021-f003:**
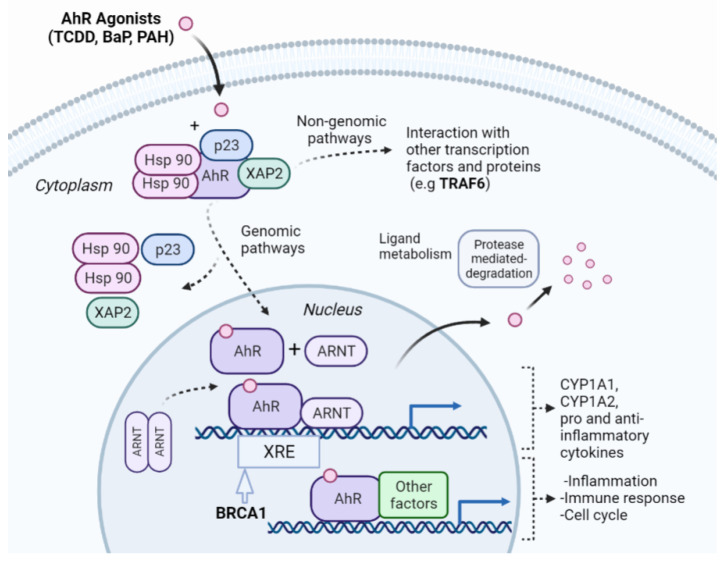
**Overview of the NR-AhR signaling pathway.** AhR is an inactive cytosolic transcription factor bound to several co-chaperones. When a ligand passively diffuses through the cell membrane and binds to AhR, the ligand-receptor complex translocates into the nucleus, and the chaperones dissociate. Once in the nucleus, the AhR dimerizes with the AhR nuclear translocator (ARNT) forming an active heterodimer. The activated heterodimer complex interacts with DNA directly or indirectly by binding to recognition sequences located in the promoter regions of target genes, such as the xenobiotic response elements (XRE). BRCA1 modulates the expression of genes involved in xenobiotic stress responses [62], and was selected by our pipeline as an important protein feature in AhR toxicity. The binding of the heterodimer complex to the DNA activates the transcription of genes leading to proteins that affect inflammation, the cell cycle, and immunological response. The toxicity of the ligand-activated AhR can also be mediated by non-genomic action through enzymatic activation and the triggering of other pathways, such as the NF-κB pathway, which involves the TRAF6 protein, another important protein selected by our pipeline. The interconnection of different pathways as shown exemplifies how our pipeline can decipher different mechanisms for AhR toxicity induced by the toxic compounds.

**Figure 4 molecules-27-03021-f004:**
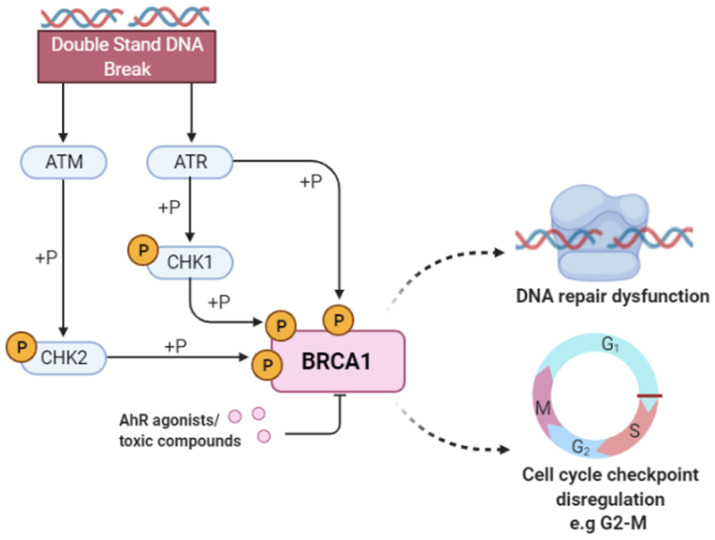
**Breast cancer type 1 susceptibility protein (BRCA1) activity in response to DNA damage.** Double-stranded breaks in DNA activate the ataxia-telangiectasia mutated (ATM) kinase and Rad3-related protein (ATR) that subsequently phosphorylate checkpoint kinase 2 (CHK2) and checkpoint kinase 1 (CHK1). The ATR/ATM-mediated phosphorylation pathway phosphorylates the BRCA1 protein and activates proteins involved in DNA damage repair and apoptosis. The binding of toxic compounds or AhR ligands leads to the suppression of the BRCA1 promoter activity, ultimately leading to increased cellular toxicity via DNA repair dysfunction and cell cycle checkpoint disregulation. Our pipeline selected BRCA1 as important for predicting AhR toxicity, along with other proteins such as RNF8, UBE2N, RAP80, NSD2, and RPA1, implying that compounds are known to induce this phenotype possibly via the modulation of proteins directly involved in DNA repair and cell cycle regulation.

**Figure 5 molecules-27-03021-f005:**
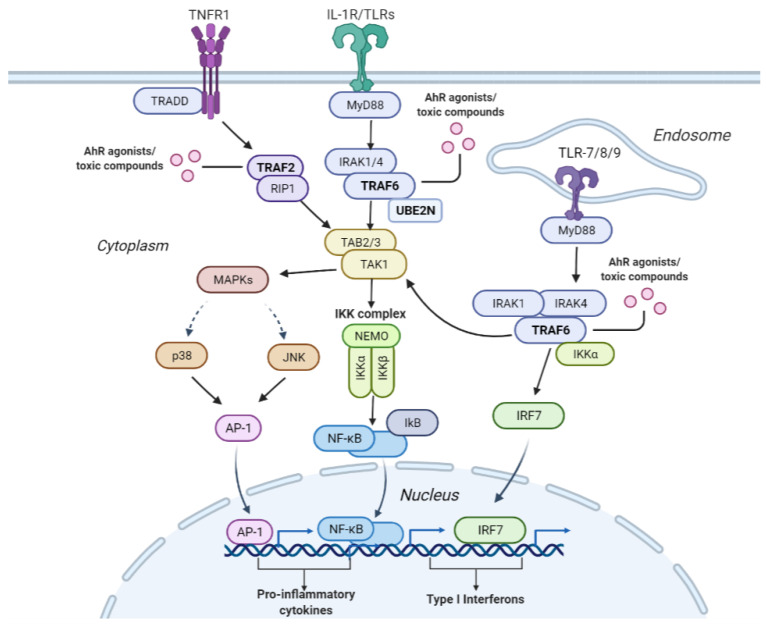
**Interconnectivity of the pathways leading to the activation of the NF-κB and MAPK signaling pathways that induce cellular inflammation.** The tumor necrosis factor receptor 1 (TNFR1) plays an essential role in pro-inflammatory activities. Upon TNFR1’s stimulation TNF receptor-associated factor 2 (TRAF2) is recruited along with receptor-interacting serine/threonine-protein kinase 1 (RIP1), which in turn activates transforming growth factor β-activated kinase 1 (TAK1). TAK1 can also be activated through the interleukin-1 receptor/toll-like receptor (IL-1R/TLR). Once IL-1R/TLR is activated, it triggers the activation of the TNF receptor-associated factor 6 (TRAF6) downstream. TRAF6 then combines with ubiquitin-conjugating enzymes Ubc13/UBE2N and ultimately activates TAK1. TAK1 then activates the nuclear factor-κB (NF-κB) signaling pathway, which recruits the NF-κB transcription factor, leading to the production of cytokines. TAK1 can also induce a cascade of mitogen-activated kinases (MAPKs), including the c-Jun kinases (JNKs) or the p38 MAPK. This activates the activator protein 1 (AP-1) transcription factor leading to the production of cytokines. In addition to activating TAK1, TRAF6 can also activate downstream interferon regulatory factor 7 (IRF7) via the TLR7/8/9-MyD88 pathway, leading to the production of type 1 interferon. TRAF6, TRAF2, and UBE2N were identified by our pipeline as important for predicting AhR toxicity, implying that these proteins may be modulated by compounds known to induce this toxic phenotype via the depicted pro-inflammatory pathways.

**Table 1 molecules-27-03021-t001:** **Details of compounds and their activity in the Tox21 dataset.** Details are given for the pre-proccessed Tox21 Data Challenge training and test sets, including the assay identifier, target, and total number of compounds assayed, as well as the partitioning of the training and test sets. We compute the imbalanced ratio (IR) for each assay, which is the ratio of the inactive non-toxic compounds to the active toxic compounds. An imbalanced ratio value closer to one signifies a fairly balanced class activity, and a value much greater than one signifies a very imbalanced class activity. The imbalanced ratio values indicate that the Tox21 dataset is highly imbalanced and that model performance can be improved by taking this into account.

In Vitro qHTS Assay Identifier	Target/Assay	Number of Compounds	Training Set			Test Set		
			Inactive	Active	IR	Inactive	Active	IR
NR-AhR	Aryl hydrocarbon receptor	7103	5777	734	7.87	521	71	7.34
NR-ER-LBD	Estrogen receptor (luciferase assay)	7509	6643	282	23.56	564	20	28.20
NR-ER	Estrogen receptor	6630	5474	651	8.41	456	49	9.31
NR-Aromatase	Aromatase	6286	5496	274	20.06	479	37	12.94
NR-PPAR-γ	Peroxisome proliferator-activated receptor γ	7039	6283	167	37.62	559	30	18.63
NR-AR	Androgen receptor	7783	6958	252	27.61	561	12	46.75
NR-AR-LBD	Androgen receptor (luciferase assay)	7298	6521	211	30.90	558	8	69.75
SR-MMP	Mitochondrial membrane potential	6316	4899	888	5.52	474	55	8.62
SR-ARE	Nuclear factor (erythroid-derived 2)-like 2 antioxidant responsive element	6339	4919	881	5.58	450	89	5.06
SR-ATAD5	Genotoxicity indicated by ATAD5	7646	6787	256	26.51	569	34	16.73
SR-p53	DNA damage p53-pathway	7358	6351	409	15.53	560	38	14.74
SR-HSE	Heat shock factor response element	7040	6144	305	20.14	574	17	33.76

**Table 2 molecules-27-03021-t002:** **Model evaluation metrics computed for the twelve Tox21 assay datasets.** For each assay, the model’s performance using F1-score, precision, recall, specificity, Mathews correlation coefficient (MCC), balanced accuracy (BA), AUROC, and AUPRC metrics is given. BA is the average of the recall and specificity and is a useful metric when evaluating imbalanced data. The overall performance of our pipeline is promising, particularly in terms of BA, depending on assays and metrics considered.

Assays	F_1_	Precision	Recall	AUCROC	AUPRC	BA	MCC	Specificity	Accuracy
NR-AhR	0.471	0.318	0.901	0.896	0.560	0.819	0.438	0.737	0.757
NR-ER-LBD	0.342	0.333	0.350	0.810	0.272	0.663	0.318	0.975	0.954
NR-ER	0.420	0.301	0.694	0.806	0.414	0.760	0.370	0.827	0.814
NR-Aromatase	0.317	0.250	0.432	0.795	0.282	0.666	0.260	0.900	0.866
NR-PPAR-γ	0.286	0.308	0.267	0.745	0.241	0.617	0.251	0.968	0.932
NR-AR	0.261	0.273	0.250	0.706	0.196	0.618	0.178	0.988	0.970
NR-AR-LBD	0.000	0.000	0.000	0.618	0.036	0.493	−0.014	0.986	0.972
SR-MMP	0.488	0.331	0.927	0.916	0.597	0.855	0.478	0.783	0.798
SR-ARE	0.425	0.305	0.697	0.757	0.403	0.692	0.294	0.687	0.688
SR-ATAD5	0.325	0.283	0.382	0.744	0.230	0.662	0.282	0.942	0.910
SR-p53	0.235	0.159	0.447	0.830	0.198	0.643	0.182	0.839	0.814
SR-HSE	0.286	0.308	0.267	0.759	0.240	0.617	0.251	0.968	0.932

**Table 3 molecules-27-03021-t003:** **Important mechanistic pathways and their proteins for the toxicity endpoint of NR-AhR.** The name of the pathway, the number of proteins present in the pathway according to Reactome, the proteins in the pathway selected by the model as important for predicting AhR toxicity, and the *p*-value from the enrichment analysis are shown. The p-value is derived using the hypergeometric distribution based on how many total important proteins were selected by the model, how many total proteins are present in the pathway, and the total number of proteins in the human proteome. The value shown is the probability of selecting at least that number of proteins present in the pathway. These results indicate that our pipeline is capable of extracting higher level biological explanations associated with AhR toxicity that have been validated via the literature.

Pathway	Total Proteins	Selected Proteins (Gene IDs)	*p*-Value
Nonhomologous End-Joining (NHEJ)	52	RNF8,UBE2N, BRCA1,NSD2	1.36 × 10${}^{-6}$
Recruitment and ATM-mediated phosphorylation of repair and signaling proteins at DNA double strand breaks	59	RNF8,UBE2N, BRCA1,NSD2	2.22 × 10${}^{-6}$
TRAF6 mediated NF-κB activation	24	TRAF2,TRAF6	2.35 × 10${}^{-6}$
DNA Double Strand Break Response	60	RNF8,UBE2N, BRCA1,NSD2	2.36 × 10${}^{-6}$
TRAF6 mediated IRF7 activation	28	TRAF2,TRAF6	3.73 × 10${}^{-6}$
Neurofascin interactions	7	NRCAM,CNTN1	5.28 × 10${}^{-6}$
DDX58/IFIH1-mediated induction of interferon-alpha/beta	77	TRAF2,RNF125, TRAF6,DDX58	6.04 × 10${}^{-6}$
RUNX3 regulates YAP1-mediated transcription	8	TEAD1,TEAD4	7.01 × 10${}^{-6}$
SUMOylation of transcription cofactors	42	RNF2,UHRF2,PIAS3	1.22 × 10${}^{-5}$
IRAK1 recruits IKK complex	14	TRAF6,UBE2N	2.21 × 10${}^{-5}$
IRAK1 recruits IKK complex upon TLR7/8 or 9 stimulation	14	TRAF6,UBE2N	2.21 × 10${}^{-5}$
YAP1- and WWTR1 (TAZ)-stimulated gene expression	14	TEAD1,TEAD4	2.21 × 10${}^{-5}$
TRAF6 mediated IRF7 activation in TLR7/8 or 9 signaling	14	TRAF6,UBE2N	2.21 × 10${}^{-5}$
TICAM1, RIP1-mediated IKK complex recruitment	19	TRAF6,UBE2N	4.05 × 10${}^{-5}$
Signal transduction by L1	20	NRP1,NCAM1	4.48 × 10${}^{-5}$
G2/M DNA damage checkpoint	78	RNF8,UBE2N,BRCA1, NSD2,RPA1	4.64 × 10${}^{-5}$
Regulation of FZD by ubiquitination	21	LRP6,LGR5	4.92 × 10${}^{-5}$
IKK complex recruitment mediated by RIP1	22	TRAF6,UBE2N	5.39 × 10${}^{-5}$
JNK (c-Jun kinases) phosphorylation and activation mediated by activated human TAK1	22	TRAF6,UBE2N	5.39 × 10${}^{-5}$
Processing of DNA double-strand break ends	81	RNF8,UBE2N,BRCA1, NSD2,RPA1	5.55 × 10${}^{-5}$
Activated TAK1 mediates p38 MAPK activation	23	TRAF6,UBE2N	5.87 × 10${}^{-5}$
Formation of Incision Complex in GG-NER	43	UBE2N,PIAS3, RBX1,RPA1	6.52 × 10${}^{-5}$
Recognition of DNA damage by PCNA-containing replication complex	31	RBX1,RPA1	1.03 × 10${}^{-5}$
TAK1 activates NFkB by phosphorylation and activation of IKKs complex	32	TRAF6,UBE2N	1.10 × 10${}^{-5}$
DNA strand elongation	32	GINS2,RPA1	1.10 × 10${}^{-5}$
Sialic acid metabolism	33	GLB1,NANP	1.16 × 10${}^{-5}$
Transcriptional Regulation by E2F6	34	RNF2,BRCA1	1.23 × 10${}^{-5}$
Negative regulators of DDX58/IFIH1 signaling	34	RNF125,DDX58	1.23 × 10${}^{-5}$
NOD1/2 Signaling Pathway	35	TRAF6,UBE2N	1.30 × 10${}^{-5}$
RUNX1 interacts with co-factors whose precise effect on RUNX1 targets is not known	36	RNF2,PCGF5	1.37 × 10${}^{-5}$
HDR through Single Strand Annealing (SSA)	37	BRCA1,RPA1	1.44 × 10${}^{-5}$
Ovarian tumor domain proteases	38	TRAF6,DDX58	1.51 × 10${}^{-5}$
Presynaptic phase of homologous DNA pairing and strand exchange	39	BRCA1,RPA1	1.58 × 10${}^{-5}$
Formation of Fibrin Clot (Clotting Cascade)	39	PROCR,GP1BB	1.58 × 10${}^{-5}$

## Data Availability

All the data and code used to pre-process the data and build the machine learning model are available at GitHub under https://github.com/lamawmouk/Tox21_FeatureSelection_SMOTEENN_RF, accessed on 2 May 2022.

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
