# Peer review of "Identifying Protein Features and Pathways Responsible for Toxicity Using Machine Learning and Tox21: Implications for Predictive Toxicology"

_molecules, 2022, doi:10.3390/molecules27093021_

Round 1

Reviewer 1 Report

The authors’ represented here an interesting manuscript “Identifying protein features and pathways responsible for toxicity using machine learning and Tox21: Implications for predictive toxicology”. From my point of view, the present study has enough merit for its publication in “Molecules” Journal

Reviewer 2 Report

In the manuscript, the authors propose a computational model for predicting the toxicity of compounds in the Tox21 dataset using the CANDO therapeutic discovery platform. Random Forest with hybrid SMOTE+ENN method were used to overcome the high imbalance of the Tox21 dataset. The resulting proteins were crossed with pathways annotated in Reactome database, and the enriched pathways were described as a case study of the NR-AhR toxicity.

The authors imply that this approach is able to extract higher-level biological explanations related to the toxicity of NR-AhR, than those from the literature. Infortunately, there is no confirmation of this statement in the text. And the claim that the authors have received more complete data than is available in the literature look strange.

Yes, the authors conducted a fairly extensive study, obtained quite interesting results, but the analysis of these results is completely unacceptable.

First of all, this concerns the Charter 2.4 Case Study for NR-AhR, which compared the results of the enrichment analysis (Table 3) with data extracted from the literature. The presented Table 3 itself is completely uninformative,

To make this work really interesting and useful for readers, especially those working in the field of wet biology, it is necessary not just to list the pathways pulled from the Reactome database in the form of a list, together with the number of proteins from Uniprote, but to conduct a real analysis of these clearly interconnected signaling pathways. Make a scheme (String-type or another design). In addition to the number of proteins from Uniprote, give at least abbreviated names, or even better, a table in the supplement with transcripts of proteins from Uniprote.

It is obvious to anyone familiar with the cellular signaling system that the information from Table 3 can be presented as a schematic map of several signaling pathways, including MARK/SAPK, NFkB, DNA-repair and so on. I believe that these data require a decent presentation. And only after this it make sense to compare them with literary data.

Of course, it would be interesting to see a similar analysis for another pathway identified by the authors related to the stress response

There are some technical problems in the design of the manuscript. Most of the tables are too wide (landscape mode) to be read correctly.

Reviewer 3 Report

Authors developed a computational model using machine learning for selecting the most important protein features for predicting the toxicity of the compounds in the Tox21 dataset using the multiscale Computational Analysis of Novel Drug Opportunities (CANDO) therapeutic discovery platform. It is useful for scientific community. But, there are many major points for addressing before publication:

  1. RF was used in this study. But authors did not provide the details of the program or package used. (see references doi: 1093/bib/bbz088, 10.1093/bib/bbaa125)
  2. The feature extraction is the important part for ML model development. But authors provide a lack information of this part.
  3. In this study, authors implemented binary classification model. Thus, there are five commonly used statistical metrics (ACC, Sn, Sp, MCC, AUC) (see references doi: 17179/excli2021-4411, 10.1038/s41598-022-08173-5, 10.17179/excli2022-4723, 10.1007/s10822-020-00368-0).
  4. For the results in Table 2, it is better if all results are represented with 3 decimal places
  5. Please carefully check the complete information in Table 1.
  6. The cross-validation test is not enough to verify the model’s effectiveness. Authors should provide the independent test results. (see references doi: 1007/s10822-021-00418-1).
  7. Since, authors employed SMOTE+ENN to solve imbalance issue, but authors did not provide the results based on the original imbalance dataset.
  8. It is better if authors compare the RF’ performance with other related ML methods, such as PLS, KNN, SVM, ETree or XGBoost etc.

Round 2

Reviewer 2 Report

The authors have responded to all my comments

Reviewer 3 Report

The current version can be accepted for publication.